# Purification of a Fc-Fusion Protein with [Bathophenathroline:metal] Complexes

**DOI:** 10.3390/antib14010011

**Published:** 2025-01-31

**Authors:** Thisara Jayawickrama Withanage, Ron Alcalay, Olga Krichevsky, Ellen Wachtel, Ohad Mazor, Guy Patchornik

**Affiliations:** 1Department of Chemical Sciences, Ariel University, Ariel 4070000, Israel; 2Israel Institute for Biological Research, Ness-Ziona 7410001, Israel; 3Faculty of Chemistry, Weizmann Institute of Science, Rehovot 7610001, Israel

**Keywords:** Fc-fusion proteins, acetylcholinesterase, non-chromatographic purification, bathophenanthroline:Zn^2+^ complexes

## Abstract

In this study, we assess an alternative Fc-fusion protein purification method that does not rely on chromatographic media or ligands. Recombinant human acetylcholinesterase, fused to the Fc domain of human IgG1 (henceforth, AChE-Fc), was purified with precipitated aromatic complexes composed of the bathophenanthroline (henceforth, batho) chelator with either Zn^2+^ or Cu^2+^ ions (i.e., [(batho)_3_:Zn^2+^] or [(batho)_2_:Cu^2+^]) in the presence of polyethylene glycol 6000 (PEG-6000). In a three-step purification process conducted at pH 7, AChE-Fc was captured by the aromatic complexes (Step 1); unbound or weakly bound protein impurities were removed with 20 mM NaCl (Step 2); and AChE-Fc was then extracted at pH 7 (Step 3) using 100 mM Na citrate buffer in 250 mM NaCl. Purified AChE-Fc was not aggregated (as determined by dynamic light scattering (DLS) and Native PAGE). However, full enzymatic activity was only preserved with the [(batho)_3_:Zn^2+^] complex. Interaction between AChE-Fc and [(batho)_3_:Zn^2+^] led to ~83–88% overall protein yield. Thirty-fold process upscaling by volume required only proportional increase in the amounts of [(batho)_3_:Zn^2+^] and PEG-6000. Efficient (95–97%) chelator recycling was achieved by recrystallization. Chelator leaching into purified AchE-Fc was estimated to be ~0.3% relative to the total amount used. Taken together, this novel procedure has the potential to provide an economical and practical avenue for the industrial purification of Fc-fusion proteins.

## 1. Introduction

Fc-fusion proteins, first introduced in 1989 [1], are recombinant products of the Fc domain of immunoglobulin G (IgG), conjugated to a ligand. Such products may be highly diverse. A wide range of ligands are used and commonly include receptor extracellular domains [2], active peptides [3], enzymes [4,5] or cytokine traps [6]. Generally, Fc-fusion proteins are obtained by conjugating the C-terminus of the ligand to the N-terminal of the flexible hinge region in the Fc domain. This fusion strategy is employed since the resulting products are stabilized by the strong interaction between the homodimeric Fc CH2-CH3 domains of IgG [7,8]. The motivation behind the use of Fc-fusion proteins in medicine is due to the observation that many potentially therapeutic proteins or peptides exhibit a short serum half-life (t_½_) which frequently reduces their practical usefulness. The short serum t_½_ is generally the direct result of degradation by endogenous peptidases or rapid renal clearance. [9] For example, efficient secretion through the kidneys is observed for macromolecules having a molecular weight of <60 KDa. [10] Fusing ligands to the Fc domain of an IgG circumvents these difficulties by increasing the size of the fusion product. Since the molecular weight of the IgG Fc domain is ~50 KDa [11], it leads to constructs that can exceed the kidney filtration cutoff (60–70 KDa) [10,12], which in turn, lengthens the Fc-fusion protein circulation time. [13,14] This is accomplished by binding to the neonatal Fc receptor (FcRn) in a pH-mediated recycling process [15,16]. The specific interaction protects Fc-fusion proteins from being degraded by lysosomes [17] and accordingly, their serum t_½_ increases in parallel with the therapeutic effect [18,19]. Inclusion of the Fc domain provides several additional advantages: (a) interaction with Fc receptors located on immune cells. This property expands the medicinal use of Fc-fusion proteins to oncology as well as vaccine production [20]; (b) increasing avidity and potency. As Fc-fusion proteins are generally homodimers, they can be polymerized into homodimer–hexamers by engineering disulfide bonds between the constant heavy domains (CH2-CH3) in the Fc domain along with the use of an extension of 18 amino acids called a tailpiece, derived from the C-terminus of immunoglobulin M (IgM) [21]). These, in turn, increase the number of binding sites, and hence, avidity [19]; (c) increasing the stability and solubility of lipophilic ligands or receptors [8,22]; and (d) improving the expression level and secretion rate of target constructs [23].

The medicinal success of Fc-fusion proteins is similar to that of intact IgG’s and is best reflected in global statistics showing that out of the ~13–16 FDA approved Fc-fusion proteins (as of 2020 [7]), five are “blockbusters” [24]. Since mAbs and Fc-fusion proteins represent the largest category of biopharmaceuticals [25], they require efficient industrial-scale production that could provide high purity products for clinical use [26]. With respect to purification, many of the same isolation methods developed for antibody purification are used for Fc-fusion proteins [26] and fall into two major categories: chromatographic and non-chromatographic.

Non-chromatographic methods may include precipitation [27], aqueous two-phase partitioning [28] and crystallization [29], but none of these have been extensively employed for Fc-fusion proteins [25]. However, affinity-based chromatographic methods that include the bacterial protein, Protein A (i.e., Protein A affinity chromatography), are primarily utilized during the initial capturing step [30]. This is commonly followed by ion exchange chromatography for the final polishing [26]. The popularity of Protein A chromatography derives from (a) high binding affinity [31] and the selectivity of the ligand toward the Fc domain via helices 1 and 2 in the three tightly packed, alpha-helical bundles present in each of the five homologous domains of Protein A [32,33,34]. This translates to high process yields for different antibody subclasses (up to 99% [30]) and purity within a single chromatographic step [25,35]; (b) the tolerance of the ligand towards acidic pH (used during the elution step), or denaturants, e.g., urea, guanidinium hydrochloride or NaOH, applied during column sanitation [36,37]; and (c) the ability to fabricate polymeric resins with high binding capacity towards target mAbs and Fc-fusion proteins [38,39].

Nevertheless, chromatographic media in which Protein A is covalently immobilized in polymethacrylate, agarose or glass beads [38,39,40] suffer from two serious drawbacks: (i) a limited flow rate caused by back pressure generated due to diffusion into the pores of the polymeric beads; and (ii) the requirement to filter samples prior to their loading on the column in order to minimize column clogging [25]. Overcoming these shortcomings has been attempted with approaches that do not rely on polymeric beads, including monolithic affinity chromatography [35,41] and specially designed membranes [42]. Though both strategies contribute to high flow rates and lower back pressures, to the best of our knowledge, they do not show sufficient binding capacity towards antibodies and Fc-fusion proteins, currently rendering these protocols impractical for industrial scale use [30,43,44].

In this study, we assess an alternative Fc-fusion protein purification method that does not rely on chromatographic media or ligands. Instead, aromatic [metal:chelator] complexes are used, composed of the commercially available bathophenanthroline chelator (henceforth, batho) and either Zn^2+^ or Cu^2+^ ions as [(batho)_3_:Zn^2+^] or [(batho)_2_:Cu^2+^]. PEG-6000 is also present. Such complexes have recently demonstrated their ability to purify human lactoferrin [45], most likely via cation-π interactions generated between the positively charged side chains of Arg and Lys amino acids of the protein and the electron cloud of the aromatic chelator. Accordingly, we have tested the ability of the above complexes to purify a recombinant homodimer, an Fc-fusion protein comprising the Fc domain of human IgG1 fused to human acetylcholinesterase (AChE) (i.e., AChE-Fc). This enzyme functions as a highly efficient bioscavenger for organophosphate nerve agents and, once fused to the Fc domain, demonstrates an extension of its serum t_½_ from minutes to 4 days (in rats) and hence, may be a practical and valuable prophylactic agent [5]. The encouraging pharmacokinetic properties of AChE-Fc, combined with its efficient neutralizing capability against diverse nerve agents and ease of production (i.e., overexpression and purification), has suggested that this fusion protein may represent a next-generation organophosphate bioscavenger. Our report aims at demonstrating the efficiency with which AChE-Fc may be isolated and purified in the absence of any chromatographic step or ligand.

## 2. Materials and Methods

### 2.1. Materials

The materials used were sodium chloride (Sigma (Rahway, NJ, USA), S7653), bathophenanthroline (GFS chemicals (Columbus, OH, USA), C038446), zinc chloride (Sigma, 208086), copper (II) sulfate pentahydrate (Sigma, 209198), polyethylene glycol-6000 (PEG-6000) (Sigma, 81260), sodium citrate tribasic dihydrate (Sigma, S4641), PBS (Bio-Lab (Lawrenceville, GA, USA), 162323G500), bovine serum albumin (BSA) (Sigma, 10775835001), 5,5-dithio-bis-(2-nitrobenzoic acid) (DTNB) (Sigma, D8130), sodium phosphate (Sigma, S9763), and acetylthiocholine iodide (ATC) (Sigma, O1480).

### 2.2. Methods

#### 2.2.1. Preparation of 0.2M Bathophenanthroline:DMSO:HCl (Batho:DMSO:HCl) Solution

Into 90 μL of dimethyl sulfoxide (DMSO) and 10 μL of 25% HCl, 6.64 mg of bathophenanthroline was added and vortexed for 5 min until total dissolution is observed. The whole procedure is conducted at 25 °C.

#### 2.2.2. Preparation of AChE-Fc Fusion Proteins

CHO-S cells that stably express AChE-Fc were cultured in serum-free CD FortiCHO cell media and supplemented with 8 mM GlutaMAX and Anti-Clumping Agent (all purchased from Gibco (Waltham, MA, USA)). Crude-culture media containing AChE-Fc were used for all further purification protocols.

#### 2.2.3. Purification of AChE-Fc via [(batho)_3_:Zn^2+^] or [(batho)_2_:Cu^2+^] Complexes

Step 1: Complex formation: the addition of 0.95 μL of [batho:DMSO:HCl] to 24 μL of 4 mM ZnCl_2_ (or 4 mM CuSO_4_) in 20 mM NaCl was followed by a 5-min incubation at 25 °C; 2.5 μL of 5M NaCl was then added and further incubated for 5 min at 25 °C. Centrifugation (21,000× *g*, 5 min at 19 °C) was applied, the supernatant discarded, and the pellet was washed once with 50 μL of cold 20 mM NaCl. Centrifugation was repeated, (21,000× *g*, 5 min at 10 °C) and the resulting pellet was used for AChE-Fc capture. Step 2: AChE-Fc capture: freshly prepared [metal:chelator] complexes were resuspended with 150 μL of overexpressed AChE-Fc (0.4 mg/mL, by enzymatic activity) plus impurities, 37 μL of 0.5 M Na citrate (pH 7) and 187 μL of 30% *w/v* of PEG-6000 in DDW. Following a 20-min incubation at 10 °C and centrifugation (21,000× *g*, 5 min at 10 °C), the supernatant was discarded, and the pellet was briefly washed with 100 μL of cold 20 mM NaCl. Step 3: AChE-Fc extraction: washed pellets were resuspended in 200 μL of 100 mM Na citrate (pH 7) and 250 mM of NaCl and incubated for 30 min at 10 °C. Centrifugation followed (21,000× *g*, 5 min, 10 °C), and aliquots from the supernatant were analyzed by SDS-PAGE.

#### 2.2.4. Dynamic Light Scattering (DLS)

AChE-Fc purified by [(batho)_3_:Zn^2+^] or [(batho)_2_:Cu^2+^] complexes or by Protein A chromatography (the control [C] sample) were dialyzed against phosphate-buffered saline (PBS) and diluted to 0.2 mg/mL. Samples were centrifuged (21,000× *g*, 10 min at 10 °C) prior to analysis, and the intensity-weighted particle size distribution of AChE-Fc was determined using the auto correlation spectroscopy protocol of the Nanophox instrument (Sympatec GmbH, Clausthal-Zellerfeld, Germany).

#### 2.2.5. Circular Dichroism (CD) Spectroscopy

Samples containing purified AChE-Fc were first dialyzed in PBS and further diluted with PBS to 0.05 mg/mL. Analysis was performed using a Chirascan CD spectrometer (Applied Photophysics(Leatherhead, UK)). CD spectra report ellipticity (θ) proportional to the difference in the absorbance of left and right circularly polarized light [θ = 3300° (AL-AR)] as a function of wavelength. A quartz cell of path length of 0.1 cm was used for the measurements. The CD spectra were recorded with a 2 nm bandwidth resolution in 1 nm steps at 25 °C. The collected CD spectra were corrected for baseline distortion by subtracting a reference spectrum of the corresponding buffer solution.

#### 2.2.6. Native PAGE

AChE-Fc samples, purified either by Protein A chromatography or via the [(batho)_3_:Zn^2+^] or [(batho)_2_:Cu^2+^] complexes, were applied to a 10% native gel, prepared according to the protocol of Trudel and Asselin [46].

#### 2.2.7. Batho Recrystallization and Quantitation

At the end of the purification protocol (as described above), pellets were incubated at 80 °C for an hour with 0.5 mL of a mixture comprising DDW/CH_3_OH (4:1, *v*/*v*) that included 250 mM EDTA at pH 8. A brief centrifugation (4 °C, RCF = 21,000, 5 min) and pellet washing with 0.5 mL DDW led to the appearance of colorless microcrystals. These were subsequently dissolved in methanol. Following the addition of 4 mM FeCl_2_ in 20 mM NaCl, the specific absorption of [(batho)_3_:Fe^2+^] at 533 nm [47] was measured. Control samples representing 100% recycling yield were prepared by mixing an identical amount of batho used for the purification protocol with the same volume of the 4 mM FeCl_2_ in 20 mM NaCl as described above. Quantitation of the red [(batho)_3_:Fe^2+^] complex was performed with a Jasco V-750 spectrophotometer (Jasco: Tokyo, Japan).

#### 2.2.8. AChE-Fc Activity

Enzymatic activity was determined according to Ellman et al., [48] in the presence of AChE-Fc substrate buffer comprising 0.1 mg/mL of bovine serum albumin (BSA), 0.3 mM 5,5-dithio-bis-(2-nitrobenzoic acid) (DTNB), 50 mM NaPi, pH 8.0 and 0.5 mM acetylthiocholine iodide (ATC) (Figure 1). Measurements were performed at 27 °C using the Thermomax ABS Plus microplate reader (Molecular Devices (New Castle, PA, USA)). Samples containing an identical amount (as determined by the ND-1000 Nanodrop, Thermo-Scientific (Waltham, MA, USA)) of either the control [C] AChE-Fc—purified via Protein A chromatography or AChE-Fc, isolated with either of the aromatic [metal:chelator] complexes as described in Section 2.2.3—were dialyzed against phosphate-buffered saline (PBS) and used to generate a series of dilutions from which the optical density at λ_max_ = 405 nm was determined following 5 min of incubation.

#### 2.2.9. Leaching Assessment

Pellets comprising [(batho)_3_:Zn^2+^] or [(batho)_2_:Cu^2+^] were prepared as described above, washed with 100 μL of 20 mM NaCl and then dissolved in 200 μL methanol. Aliquots from the fully dissolved complexes were diluted with methanol, and the intensity at 280 nm was measured with a UV–Vis (Jasco V-750) spectrophotometer. The values obtained, with a known dilution factor, represented the total amount (i.e., 100%) of the [metal:chelator] complexes initially present in the system and prior to the capturing step. The protocol was repeated in the presence of 100 mM Na Citrate (pH 7), 250 mM NaCl.

#### 2.2.10. Scanning Electron Microscopy (SEM)

Freshly prepared and precipitated [(batho)_3_:Zn^2+^] or [(batho)_2_:Cu^2+^] complex samples in 20 mM NaCl were loaded onto a carbon-coated copper grid Type -B (200 mesh) and dried overnight in a desiccator at 25 °C. Images were obtained using the UHR-MAIA3 TESCAN SEM with an In-Beam SE detector at HV 25 kV and magnification, 500 kX.

## 3. Results and Discussion

The focus on two different bathophenanthroline (batho) complexes (i.e., [(batho_)3_:Zn^2+^] or [(batho)_2_:Cu^2+^]) (Figure 2) was motivated by our intention to study whether the metal coordination number can affect the process yield and/or purity. Since Zn^2+^ [49] and Cu^2+^ [50] form [1:3] and [1:2] complexes, respectively, with batho, and Zn^2+^ is less toxic than Cu^2+^ [51], they appear to represent suitable models. The use of batho as the preferred chelator derives from a previous study demonstrating its superiority with respect to the purity of the recovered target protein when compared to other hydrophobic chelators [45] combined with its ability to be recycled rapidly, and essentially quantitatively, by recrystallization [52].

### 3.1. Precipitation of the Metal-Chelator Complexes

Both [(batho)_3_:Zn^2+^] and [(batho)_2_:Cu^2+^]) precipitated upon the addition of 0.5M NaCl. Scanning electron microscopy (SEM) showed that these precipitates do not contain crystals; rather, they consist of micron-sized, entangled and porous meshes, independent of the metal used (Figure 2). They share some morphological resemblance to the matrix of monolithic chromatography [41]. Interestingly, cation identity (i.e., Zn^2+^ or Cu^2+^) had no marked effect on the morphology of the resulting precipitates. Incubation at 10 °C of the overexpressed human AChE-Fc plus cellular impurities with either of the resuspended complexes led to quantitative capture when 50 mM NaCitrate (pH 7) was used as a buffer whereas other buffers tested (e.g., Tris or NaPi at the same pH) were less efficient. Target extraction from the precipitated complexes was achieved at 10 °C with 100 mM NaCitrate (pH 7) in 250 mM NaCl. Under these conditions, a relatively pure target was recovered (>95%, by SDS-PAGE) (Figure 3A,B, lanes 3–6). Band migration of the Fc-fusion protein under denaturing and reducing conditions was very similar to that observed earlier [5], and process reproducibility was high; no significant change in target band intensity was observed in four independent purification trials performed on four different days (Figure 3A,B, lanes 3–6). Thus, the aromatic chelator complexes demonstrated their ability to purify an Fc-fusion protein in the absence of a specific chromatographic ligand.

### 3.2. The Role of PEG 6000

We found that in addition to the required presence of batho, metal cation and the NaCitrate buffer (pH 7), the presence of 15% *w/v* PEG-6000 is also essential, as at 13% *w/v*, a reduction in overall process yield was already observed, and at 9% *w/v*, only trace amounts of the target fusion protein were recovered (Appendix A, lanes 4 and 6). This requirement may derive from the inherent property of polyethylene glycols (PEGs) as osmotically active polymers, promoting molecular crowding with parallel preservation of enzymatic activity [53,54,55,56]. We therefore suggest that the presence of 15% *w/v* PEG-6000 during the capturing step brings the fusion proteins into close proximity with the porous, entangled meshes of the precipitated aromatic complexes that, in turn, promote binding between the two.

### 3.3. Aggregational State of the Recovered Fc-AChE

In order to be approved by the American Food and Drug Administration, biopharmaceuticals must be shown to not have undergone self-association. It was therefore essential to assess whether the purified Fc-fusion protein is monodisperse [57]. We performed dynamic light scattering (DLS) measurements that are commonly employed to determine protein hydrodynamic size distributions [58]. The results showed that the hydrodynamic particle size (43 nm) of AChE-Fc that served as the control **[C]** sample, i.e., isolated with Protein A chromatography, was essentially identical to the particle sizes (42–40 nm) of AChE-Fc that had been subjected to purification by either the [(batho)_3_:Zn^2+^] or [(batho)_2_:Cu^2+^]) complexes, respectively (Figure 4A).

To further validate protein monodispersity, the purified Fc-fusion protein was analyzed by native polyacrylamide gel electrophoresis (Native PAGE). For Native PAGE, neither detergent (i.e., sodium dodecyl sulfate, SDS) nor reducing agents (e.g., β-mercaptoethanol) are present. Under these non-denaturing and non-reducing conditions, we found that all protein samples, including the control sample, migrated a distance on the gel consistent with molecular weight in the range of 180–220 KDa (Figure 4B lanes 2–4). These results confirmed that all three samples contained a relatively homogenous population of fusion proteins regardless of the purification strategy used (i.e., Protein A chromatography or aromatic [metal:chelator] complexes + PEG-6000).

### 3.4. Enzymatic Activity

Preservation of enzymatic activity provides important verification of the native state of an enzyme. The enzymatic activity of acetylcholinesterase prior to and following protein purification was compared using Ellman’s assay [59] (Figure 1). While the use of the [(batho)_3_:Zn^2+^] complex fully preserved catalytic activity when compared to the target purified via Protein A chromatography, the analogous [(batho)_2_:Cu^2+^] complex reduced enzymatic activity by ~50% (Figure 5A). As the only difference between the aromatic complexes is cation identity, this implies that Cu^2+^ ions must be responsible for the reduction in enzymatic activity. We suggest a possible explanation: the much lower binding affinity of the 1,10-phenenthroline moiety to Cu^2+^ (log K_eq_ = 10.69) vs. Zn^2+^ (log K_eq_ = 24.3) at 25 °C [60] translates to a greater concentration of free Cu^2+^ ions in the solution. These, in turn, may bind non-specifically to side chains of acetylcholinesterase, thereby affecting its activity. Preservation of enzymatic activity was compared with circular dichroism (CD) spectroscopy. CD is a widely used, non-invasive analytical tool capable of assessing secondary structural alterations in proteins [61]. The positive ellipticity peak maximum at 205 nm and the negative peak centered at 218 nm were displayed by the control [C] sample, containing Fc-AChE purified with Protein A, and by the recovered target purified with either of the [metal:chelator] complexes, thereby indicating no significant alteration in protein secondary structure (Figure 5B). Importantly, the 218 nm peak is characteristic of an anti-parallel, β-pleated sheet, which is the dominant antibody secondary structural motif [62]. Here, the source must be the Fc domain of the target, since AChE is characterized by an α/β-hydrolase fold consisting of a central β-sheet packed between two layers of α-helices [63]. A recent extensive study of mAbs CD spectra [64] demonstrated that there is indeed significant heterogeneity in these spectra, in part possibly due to differences in glycosylation patterns [65] but also due to exposure to low pH. We suggest that the significantly lower peak amplitudes of the control [C] sample in Figure 5B may be due to exposure to acidic conditions during Protein A chromatography.

### 3.5. Binding Interactions and Process Yield

Since preservation of enzymatic activity was observed only with the [(batho)_3_:Zn^2+^] complex, we proceeded to focus on this complex and used Ellman’s colorimetric assay as a tool for quantitating process yield (Figure 1). We found that (i) 98% of the Fc-AChE molecules bind to the [(batho)_3_:Zn^2+^] complex during the capturing step; (ii) ~5% are lost during the washing step (iii); and an overall yield of 83% is obtained at the end of the process following the extraction step. The fact that 98% of the Fc-fusion protein molecules are bound to the aromatic [(batho)_3_:Zn^2+^] complex during the capturing step suggests the existence of high binding affinity between the two.

This observation initiated an attempt to identify the nature of the interactions responsible for such tight binding. We introduced free amino acid monomers during either the capturing step or during the extraction step which might compete with the side chains of the Fc-fusion protein for (1) binding to the aromatic [(batho)_3_:Zn^2+^] complex or (2) for being extracted from the precipitate into the supernatant. Competition upon protein binding or extraction would translate into lower overall process yield that could be quantitated by SDS-PAGE. Gels are presented in the Appendix A. We note that acetylcholinesterase enzymes are characterized by a high net negative charge and by a non-uniform surface charge distribution, giving rise to a negative electrostatic potential extending over most of the molecular surface. On the other hand, the overall surface charge of IgG is almost neutral at pH 7. We suggest that it is the Fc domain, rather than AChE, that is more likely to form chelation interactions during the capturing step with the [(batho)_3_:Zn^2+^] amphiphilic complex, attributed to an excess of imidazole (His) and carboxylate (Glu) side chains. In addition, Fc can participate in a large number of Van der Waals (induced dipole–dipole) interactions due to its Val and Pro residues.

### 3.6. Binding Capacity

The maximal binding capacity of the [(batho)_3_:Zn^2+^] complex for the Fc-AChE target was studied by repeating the purification process with a constant amount of [(batho)_3_:Zn^2+^] accompanied by a step-wise increase of the target fusion protein (Appendix A). Purity was essentially preserved when the amount of AChE-Fc + impurity proteins was doubled (Appendix A, lane 3 vs. lane 4), but at larger volumes, a small decrease in purity of AChE-Fc was indeed observed, perhaps due to the larger amount of impurities present in the system without the parallel increase of the [(batho)_3_:Zn^2+^] complex (Appendix A, lane 3 vs. lanes 5–6). Process yields, however, improved from 82% to 95% and 98% when the volume of the target protein plus impurities was increased from 150 µL (as described in the Methods section) to 300 µL and 450 µL, respectively, and decreased at 600 µL (Appendix A, lanes 3 to 6). Since 300 µL of a sample of AChE-Fc + impurities contains 120 µgr of the target (by enzymatic activity) and requires ~65 µgr of batho to reach a relatively pure Fc-fusion protein (Appendix A, lane 4), we conclude that 1 g of AChE-Fc can be isolated with ~0.55 g of batho. Under these working conditions, the molar ratio between batho (MW = 332) and AChE-Fc (MW ~240 KDa) is ~375:1, respectively. These findings are encouraging: a very limited mass of chelator is required to efficiently purify a much larger mass of AChE-Fc.

### 3.7. Process Upscaling

The impact of reaction volume on process efficiency was also evaluated. We found that a 30-fold increase from 0.375 mL to 11 mL required only a proportional increase in the [(batho)_3_:Zn^2+^] complex and PEG-6000 (Figure 6). As no reagent adjustments were needed, it may represent a good starting point for further process upscaling.

### 3.8. Chelator Recycling

With respect to process economics (as well as to environmental safety), recycling the hydrophobic chelator was considered to be essential; it represents ~85% of all raw material costs. We therefore aimed at developing a rapid, non-chromatographic, cost-effective and simple-to-implement protocol for chelator recycling. We exploited the extremely low water solubility of the batho chelator combined with its fully aromatic structure. These two properties suggested that it would rapidly crystallize from aqueous media due to its highly hydrophobic nature and ability to form π-π interactions acting as stable nucleation centers to support rapid crystal growth. Accordingly, the pellets that remained at the end of the purification process were incubated for an hour at 80 °C with 0.5 mL of a mixture containing DDW/CH_3_OH (4:1, *v*/*v*) that included 250 mM EDTA, pH 8 (Figure 7A). These conditions were expected to dissolve the [(batho)_3_:Zn^2+^] complex, as previously demonstrated for the analogous [(batho)_3_:Fe^2+^] complex, [52]. This allowed the excess EDTA molecules to compete with batho for Zn^2+^ binding and to suppress reforming of the [(batho)_3_:Zn^2+^] complex. Indeed, colorless crystals were observed following the gradual cooling of the medium from 80 °C to 4 °C and washing of the resulting pellet with DDW (Figure 7A). Mass spectrometry (Figure 7B) and HPLC analysis indicated that the micro-crystals are composed of 97% pure batho (Figure 7B,C), a similar purity level to that determined for pristine, commercial batho crystals. The recycling yield was found to be close to quantitative at (95–97%). These values are obtained by measuring the specific absorption of the [(batho)_3_:Fe^2+^] at 533 nm [47] after dissolving the entire amount of recycled chelator with methanol, adding 4 mM FeCl_2_ in 20 mM NaCl and comparing the measured OD values to those obtained from the total quantity of batho present initially in the system (Figure 7D).

### 3.9. Estimated Chelator Leaching Under Extraction Conditions

Since batho is aromatic, we quantitated chelator leaching by its absorption at 280 nm under extraction conditions of 100 mM NaCitrate, 250 mM NaCl and pH 7. A freshly precipitated [(batho)_3_:Zn^2+^] complex was prepared as described in the Methods section, and the assessment of chelator leaching was determined under standard extraction conditions. Batho leaching was found to be insignificant and dependent on temperature. Chelator leaching at 10 °C and at 25 °C was in the range of 0.27–0.33% and 0.31–0.38% respectively, relative to the total amount of chelator present.

## 4. Conclusions

An Fc-fusion protein, AChE-Fc, was efficiently purified under mild conditions—neutral pH, 10 °C—without the need for a chromatographic step, specific ligand or resin. We used a precipitated aromatic chelator [(batho)_3_:Zn^2+^] complex combined with PEG-6000, thereby eliminating the surfactants which played a role in our previous antibody purification protocols [66]. Preservation of enzymatic activity of the fusion partner, was combined with the high binding capacity of the [(batho)_3_:Zn^2+^] complex. The maintenance of the process yield and purity even upon a 30-fold increase in reaction volume suggests that the purification process studied may be readily applied to other Fc-fusion proteins and, in particular, to those that are acid labile. Purification is rapid—less than one hour—while requiring mass fraction chelator of one-half relative to the amount of the target Fc-fusion protein. Chelator leaching into the purified protein target is estimated to be only ~0.27% relative to the total amount of chelator present. There was no loss of yield or purity upon a 30-fold increase in process volume, and only a proportional increase in reagents was required. The large pore sizes of the precipitated [(batho)_3_:Zn^2+^] complex (20–80 nm) should allow the rapid diffusion of large Fc-fusion proteins similar to that achieved with porous monolithic affinity chromatography supports. Efficient chelator recycling is achieved by a rapid and cost-effective crystallization protocol. All these positive attributes suggest that the procedure described above has the potential to provide an economical and practical avenue for the industrial purification of Fc-fusion proteins.

## Figures and Tables

**Figure 1 antibodies-14-00011-f001:**
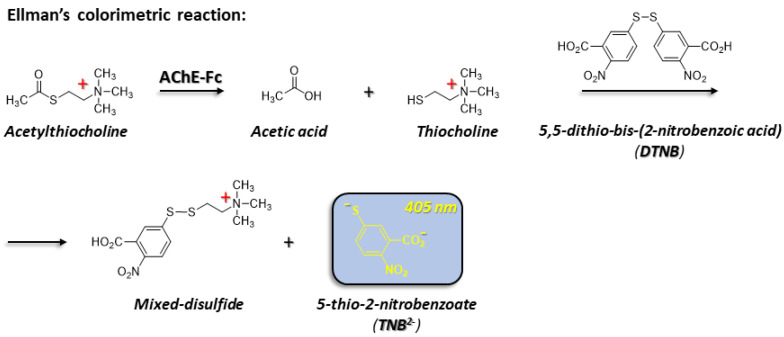
Ellman’s activity assay [48] as applied to the Fc-AChE target sample.

**Figure 2 antibodies-14-00011-f002:**
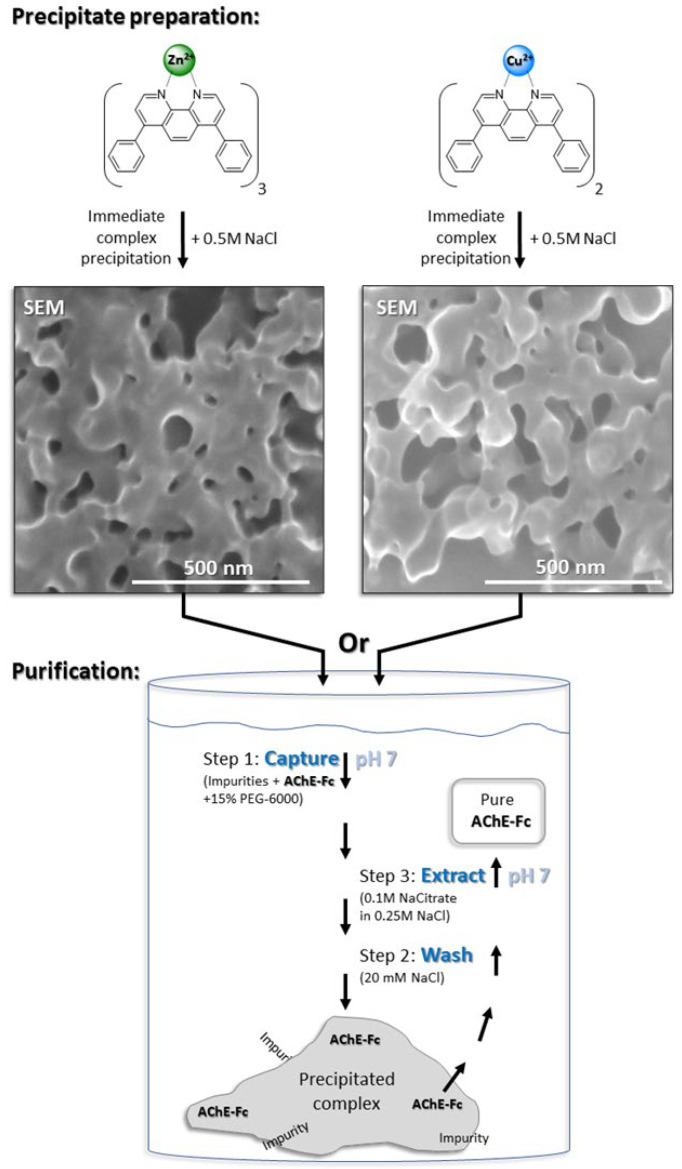
Illustration of the three-step purification protocol for AChE-Fc using [(batho):cation] complexes and 15% w/v PEG-6000. The aromatic chelator complexes are rapidly precipitated by 0.5M NaCl and then washed with 20 mM NaCl. In scanning electron microscope (SEM) imaging, the sedimented complexes appear as porous, irregular meshes. The overexpressed AChE-Fc-fusion protein is captured at pH 7 by the washed pellet and 15%w/v PEG-6000 (Step 1). Weakly bound proteins are removed with 20 mM NaCl (Step 2), and the fusion protein is extracted after 30 min incubation at 10 °C in the presence of 100 mM Na citrate (pH 7) and 250 mM NaCl (Step 3).

**Figure 3 antibodies-14-00011-f003:**
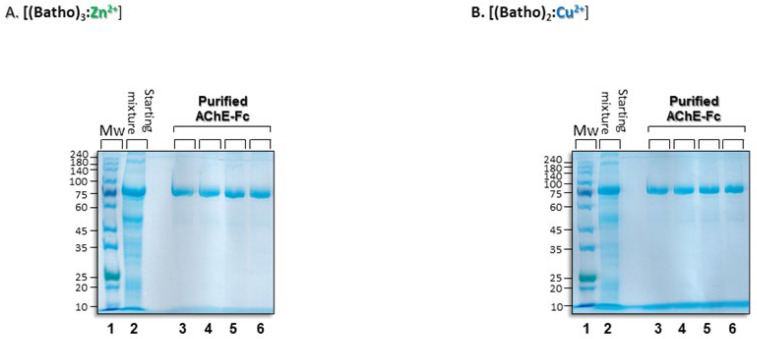
SDS-PAGE gels (with β-mercaptoethanol) of AChE-Fc homodimers with aromatic [(batho)_3_:Zn^2+^] or [(batho)_2_:Cu^2+^] complexes and 15% w/v PEG-6000. A. Lane 1: molecular weight markers; lane 2: total amount of recombinant human AChE-Fc plus impurities, prepared according to Ref. [5] and added to each purification trial; lanes 3–6: recovered AChE-Fc: the capture and washing steps (as described in the Methods section), followed by extraction at 10 °C for 30 min in the presence of 100 mM Na citrate (pH 7) and 250 mM NaCl. (**B**). As in (**A**), but with Cu^2+^ ions. Gels are Coomassie stained.

**Figure 4 antibodies-14-00011-f004:**
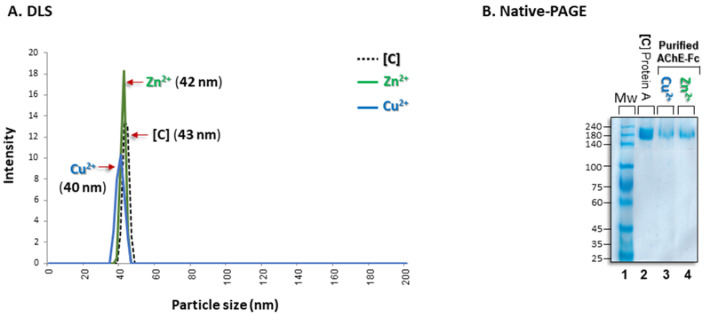
Dynamic light scattering (DLS). (**A**). Hydrodynamic particle size distribution of human AChE-Fc purified with pellets composed of either [(batho)_3_:Zn^2+^] or [(batho)_2_:Cu^2+^] complexes (as described in the Methods section) or with Protein A chromatography [C]. In all cases, purified human AChE-Fc was diluted with PBS to 0.2 mg/mL and analyzed at 25 °C. (**B**). Native PAGE gel electrophoresis. Lane 1: molecular weight markers; lane 2: AChE-Fc (5 µgr) purified by Protein A chromatography; lanes 3–4: AChE-Fc (5 µgr) purified by [(batho)_3_:Zn^2+^] or [(batho)_2_:Cu^2+^] complexes, respectively, as described in the Experimental section. Gels are Coomassie stained.

**Figure 5 antibodies-14-00011-f005:**
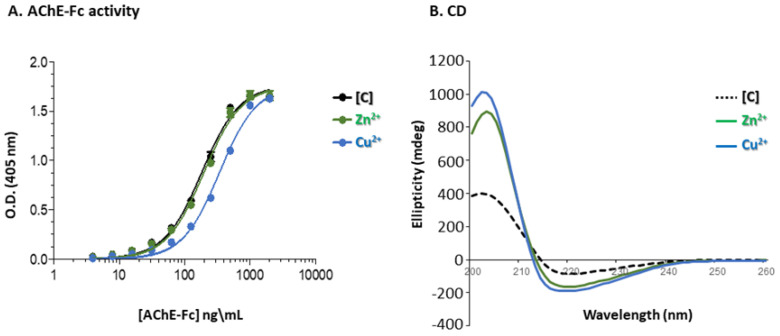
(**A**). Acetylcholinesterase activity measured by Ellman’s activity assay (see Figure 1) obtained for AChE-Fc purified via Protein A chromatography [C] (black line), [(batho)_3_:Zn^2+^] (green line) and [(batho)_2_:Cu^2+^] (blue line). (**B**). Circular dichroism (CD) spectroscopy. Far UV CD spectra of AChE-Fc purified by Protein A chromatography [C] (black dotted line) is compared to AChE-Fc isolated with either the [(batho)_3_:Zn^2+^] complex (green line) or [(batho)_2_:Cu^2+^] complex (blue line). Protein samples were diluted with PBS to 0.05 mg/mL.

**Figure 6 antibodies-14-00011-f006:**
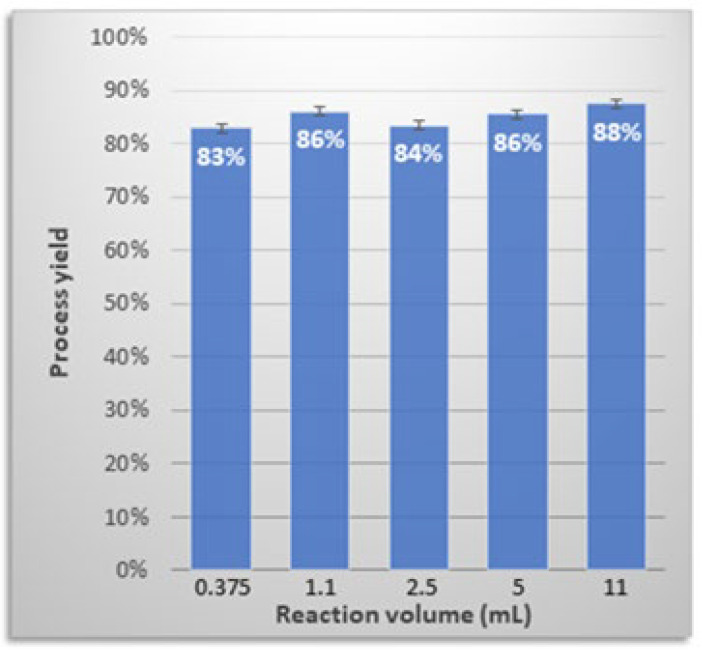
Process efficiency as a function of purification volume using the [(batho)_3_:Zn^2+^] complex. Process yields were determined by densitometry using ImageJ (NIH) and are based on at least three independent experiments conducted on different days.

**Figure 7 antibodies-14-00011-f007:**
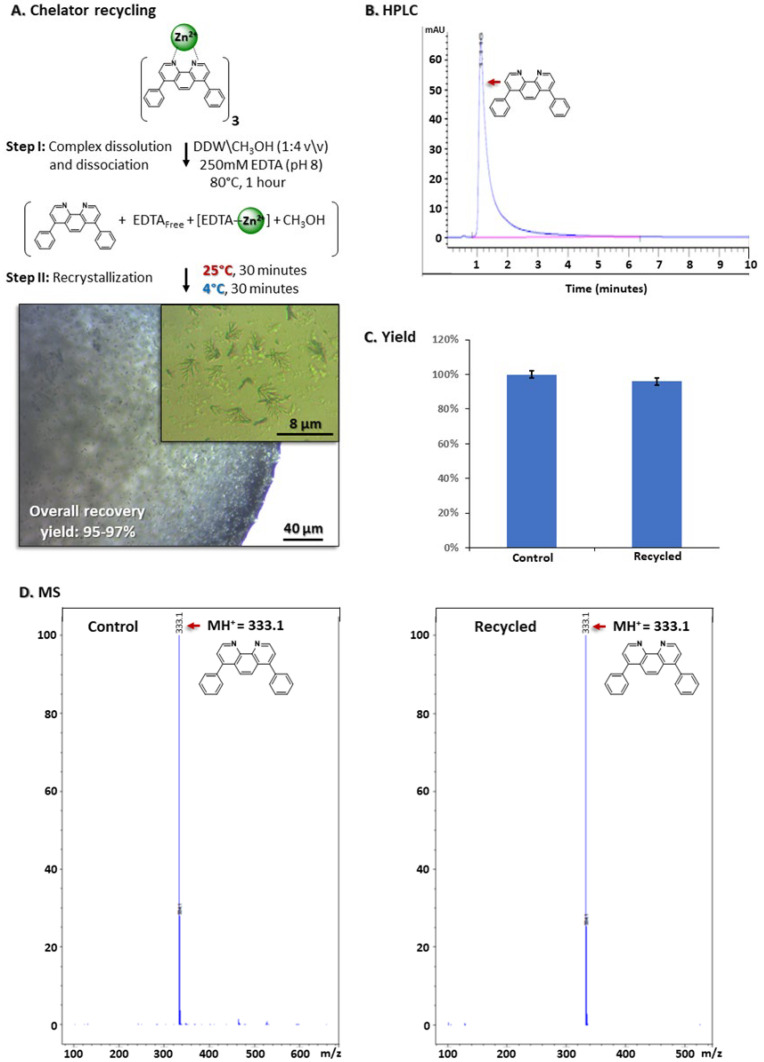
Batho recycling via recrystallization. (**A**). Illustration of the process leading to microcrystals of the recycled chelator from its complex with zinc ions (i.e., [(batho)_3_:Zn^2+^] at the end of the purification protocol. Pellets containing the [(batho)_3_:Zn^2+^] complex were dissolved in DDW/methanol (4:1 *v*/*v*) containing 250 mM EDTA (pH 8), 25 °C and incubated for 1 h in an open Eppendorf tube to allow slow evaporation of methanol. Further incubation at indicated temperatures was followed by centrifugation for 5 min at 4 °C, at relative centrifugal force (RCF) = 21,000. The supernatant was then discarded. The resulting pellet was washed twice with DDW: centrifugation for 5 min at 4 °C was then applied, (RCF = 21,000). Light microscopy images of the recycled microcrystals are shown. (**B**). HPLC analysis. (**C**). Process yield calculated by specific absorption of the recycled chelator with Fe^2+^ at 533 nm (as described in the Methods section). (**D**). Mass spectrometry of the pure, commercial chelator [C] and the recycled chelator.

## Data Availability

The data presented in this study are available on request from the corresponding author.

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
