# Peer review of "Purification of a Fc-Fusion Protein with [Bathophenathroline:metal] Complexes"

_2073-4468, 2025, doi:10.3390/antib14010011_

Round 1
Reviewer 1 Report
Comments and Suggestions for Authors
The authors of this manuscript utilized [bathophenanthroline:metal] complexes for AChE-Fc purification and found that a 30-fold increase in the fusion protein required only a proportional increase in the [(bath)3:Zn2+] complex, with no need for adjustments to other reagents. This could serve as an alternative method for industrial-scale purification, offering an economical and practical avenue for purifying fusion proteins.
I would suggest that the authors address the following concerns:
1. In lines 220-221, the authors indicate that (bath)3:Zn2+ is less toxic than (bath)3:Cu2+. Therefore, evaluating the potential toxicity of the metal ions and chelators for therapeutic applications is essential.
2. The environmental impact of the solvents and metal ions used in the purification process should also be considered, as this could raise concerns for large-scale applications.
3. The authors stated that the protein concentration used for CD spectroscopy was 0.2 mg/mL for all samples. Why do the CD spectra intensities differ in Figure 5B? I suggest the authors redo the CD analysis using constant protein concentrations throughout the experiment.
4. Minor revisions in English are necessary. For example:
In line 35, correct the spelling “immunoglobulin”
Line 92, correct the spelling “chromatography”
Line 111, correct the spelling “capability.”
Line 130, use the correct format for 25oC; 25°C
Line 134, adjust to ZnCl2, CuSO4; ZnCl2, CuSO4.
Line 244, remove the duplicate “the”
Line 358, correct the spelling “molecules”
Author Response
Comment 1. In lines 220-221, the authors indicate that [(bath)3:Zn2+] is less toxic than [(bath)2:Cu2+]. Therefore, evaluating the potential toxicity of the metal ions and chelators for therapeutic applications is essential.
Response:
Both Zn2+ and Cu2+ are considered to be essential micronutrients, while presenting widely differing physiological toxicity at intake levels exceeding recommended amounts. As a standard requirement of the American Food and Drug Administration for new treatment approval, a pharmaceutical company using our purification/separation procedure would of course conduct a comprehensive study to assess oral/intravenous intake patient safety.
Comment 2. The environmental impact of the solvents and metal ions used in the purification process should also be considered, as this could raise concerns for large-scale applications.
Response:
Our motivation for studying the recrystallization of the aromatic bathophenanthroline chelator was initially economical. However, it is also clear that the ability to recycle the chelator is also advantageous from the point of view of environmental safety. We have added a (highlighted) comment to this effect in Section.3.8. With respect to divalent cations, it is well-known that copper pollution in wastewater, at levels ranging from 2.5 mg/L to 10,000 mg/L, poses a threat both to human health as well as to the environment. A review of technological advances for removing copper ions from industrial waste has recently been published: https://doi.org/10.3390/ijerph20053885. There are also several approved methods of disposing of organic solvents on an industrial scale; however, we have no experience with them, even on a laboratory scale.
Comment 3. The authors stated that the protein concentration used for CD spectroscopy was 0.2 mg/mL for all samples. Why do the CD spectra intensities differ in Figure 5B? I suggest the authors redo the CD analysis using constant protein concentrations throughout the experiment.
Response:
Although differences in protein concentration can indeed give rise to different peak amplitudes in CD spectra, as noted in the caption to Figure 5B, the same protein concentration was used ( 0.05mg/mL) for the spectra of each of the three samples. To provide alternative explanations, we have added the following text (highlighted) to the revised manuscript: A recent extensive study of mAbs CD spectra {Bruque, M.G. et al., Analysis of the Structure of 14 Therapeutic Antibodies Using Circular Dichroism Spectroscopy, Anal. Chem., 2024, 96: 15151-15159} demonstrated that there is indeed significant heterogeneity in these spectra, in part possibly due to differences in glycosylation patterns, {Higel F. et al., N-glycosylation heterogeneity and the influence on structure, function and pharmacokinetics of monoclonal antibodies and Fc fusion proteins, European Journal of Pharmaceutics and Biopharmaceutics 2016,100: 94-100} but also to exposure to low pH. We suggest that the significantly lower peak amplitudes of the control [C] sample in Figure 5B may be due to exposure to acidic conditions during Protein A chromatography.
Comment 4. Minor revisions in English are necessary. For example:
In line 35, correct the spelling “immunoglobulin” - Corrected
Line 92, correct the spelling “chromatography” - Corrected
Line 111, correct the spelling “capability.” - Corrected
Line 130, use the correct format for 25oC; 25°C - Corrected
Line 134, adjust to ZnCl2, CuSO4; ZnCl2, CuSO4. - Corrected
Line 244, remove the duplicate “the” - Corrected
Line 358, correct the spelling “molecules” - Corrected
Response: All spelling, format and grammatical errors have been corrected and highlighted in the revised manuscript.

Reviewer 2 Report
Comments and Suggestions for Authors
The manuscript presents an interesting approach to purifying Fc-fusion proteins using a novel method that avoids traditional chromatographic techniques. The authors focus on the utilization of bathophenanthroline-metal complexes, particularly with zinc and copper, combined with polyethylene glycol (PEG) as a precipitating agent. Overall, the study provides valuable insights into a potentially cost-effective purification strategy, which is essential for the industrial application of biopharmaceuticals.
Specific comments:
1, how were Fc-fusion proteins produced? Or how were the crude solutions of Fc-fusion proteins prepared ? these should be detailed in Materials and Methods.
2, why was kjeldahl determination not used for the calculation of process yield?
3, why was activity yield not used for the determination of process yield?
4, HPLC purification of Fc-fusion proteins should be compared to the developed complexes. The economic costs of the two methods should be compared.
5, there are minor typographical and grammatical errors throughout the text. A thorough proofreading is necessary for improved clarity and professionalism.
Comments on the Quality of English Languagethere are minor typographical and grammatical errors throughout the text. A thorough proofreading is necessary for improved clarity and professionalism.
Author Response
Comment 1. How were Fc-fusion proteins produced? Or how were the crude solutions of Fc-fusion proteins prepared? these should be detailed in Materials and Methods.
Response:
CHO-S cells that stably express AChE-Fc were cultured in serum-free CD FortiCHO cell media, supplemented with 8 mM GlutaMAX and Anti-Clumping Agent (all purchased from Gibco). Crude culture media containing AChE-Fc was used in all further downstream purification. These details have been added to the Materials and Methods section, and are now highlighted.
Comment 2. Why was kjeldahl determination not used for the calculation of process yield?
Response:
The kjeldahl method can indeed be used to measure protein content. However, the primary focus of this study was to develop a novel, delicate and cost-effective purification method that retains the activity of sensitive enzymes. We therefore concluded that it is more correct to directly follow the AChE enzymatic activity; otherwise, measuring total protein content might result in overestimation of the purification efficacy. For example, please note (Figure 4A) that when the [(batho)2:Cu2+] complex was used in the current study, non-active enzyme was indeed present in the sample .
Comment 3. Why was activity yield not used for the determination of process yield?
Response:
Process yield and the loss of Fc-fusion protein during the "washing step" were in fact both determined by the enzymatic activity of acetylcholinesterase as noted in lines 371-374 of the revised version of the manuscript. We reported that 98% of the Fc-AChE molecules bind to the [(batho)3:Zn2+] complex during the capturing-step; (ii) ~5% are lost during the washing-step and (iii) an overall yield of 83% is obtained following the extraction-step.
Comment 4. HPLC purification of Fc-fusion proteins should be compared to the developed complexes. The economic costs of the two methods should be compared.
Response:
An economic comparison between Protein A chromatography (HPLC is generally not used for protein macromolecules) and our chelator-based purification strategy for fusion proteins is of course required. The high cost of Protein A resins and the limited binding capacity of Protein A affinity columns for high antibody concentrations, both of which are problematic in an industrial setting, as well as the high maintenance costs associated with the use of liquid chromatography instrumentation, were among the motivating factors for our development of the cost-effective strategy presented in our manuscript. However, the reaction volumes with which we are currently able to work are certainly too far from an industrial scale to make any economic assessment at all meaningful.
Comment 5. There are minor typographical and grammatical errors throughout the text. A thorough proofreading is necessary for improved clarity and professionalism.
Response:
In the revised version of our manuscript, we have corrected all typographical, spelling and grammatical errors. These corrections are now highlighted.

Round 2
Reviewer 1 Report
Comments and Suggestions for Authors
The authors addressed the previous comments well enough.
Reviewer 2 Report
Comments and Suggestions for Authors
it is acceptable